# Age and structure of a model vapour-deposited glass

Daniel R. Reid[1], Ivan Lyubimov[1], M.D. Ediger[2] & Juan J. de Pablo[1,3]

Glass films prepared by a process of physical vapour deposition have been shown to have thermodynamic and kinetic stability comparable to those of ordinary glasses aged for thousands of years. A central question in the study of vapour-deposited glasses, particularly in light of new knowledge regarding anisotropy in these materials, is whether the ultra-stable glassy films formed by vapour deposition are ever equivalent to those obtained by liquid cooling. Here we present a computational study of vapour deposition for a two-dimensional glass forming liquid using a methodology, which closely mimics experiment. We find that for the model considered here, structures that arise in vapour-deposited materials are statistically identical to those observed in ordinary glasses, provided the two are compared at the same inherent structure energy. We also find that newly deposited hot molecules produce cascades of hot particles that propagate far into the film, possibly influencing the relaxation of the material.

[1] Institute for Molecular Engineering, University of Chicago, 5640 South Ellis Avenue, Chicago, Illinois 60637, USA. [2] Department of Chemistry, University of Wisconsin-Madison, Madison, Wisconsin 53706, USA. [3] Institute for Molecular Engineering, Argonne National Laboratory, 9700 Cass Ave, Lemont, Illinois 60439, USA. Correspondence and requests for materials should be addressed to D.R.R. (email: danielreid@uchicago.edu) or to I.L. (email: ilyubimo@uchicago.edu) or to M.D.E. (email: ediger@chem.wisc.edu) or to J.J.d.P. (email: depablo@uchicago.edu).

Glasses represent kinetically arrested states of matter, whose characteristics depend strongly on the process of formation[1]. They are generally prepared by gradual cooling of a liquid to temperatures below the glass transition, $T_g$, of the corresponding bulk material. The properties of liquid-cooled, 'ordinary' glasses depend on cooling rate and on the 'age' of the glass—the amount of time that the material is allowed to rest at a given temperature (below $T_g$). Lower cooling rates (or ageing) lead to materials that lie deeper in the underlying potential energy landscape. They tend to have a higher density[2,3], greater mechanical strength[4], lower enthalpy[2] and higher onset temperature (the temperature at which the film transforms from a glass into a liquid upon heating)[5], than those prepared by fast cooling. Higher stability is desirable in a wide range of applications, from organic electronics[6] to drug delivery[7].

Recent experimental work has shown that glasses prepared by a process of physical vapour deposition (PVD) can reach levels of stability that are equivalent to those of liquid-cooled glasses allowed to age for thousands of years[3,8]. These highly stable PVD glasses are formed by depositing the glass former onto a substrate whose temperature is somewhat lower than $T_g$. It has been proposed that newly deposited molecules can freely explore configurational space near the surface of the growing film[9,10], leading to molecular arrangements that correspond to lower free energy states than those accessible by quenching a bulk liquid[3].

The properties of three-dimensional (3D) PVD glasses have also been examined in computer simulations. On the one hand, results for a 3D model glass former consisting of a binary mixture of spherical particles indicate that vapour deposition leads to materials that exhibit higher kinetic stability, and whose structure is similar to that of their liquid-cooled counterparts[11]. On the other hand, simulations of model glasses consisting of anisotropic molecules suggest that a PVD process leads to materials that exhibit varying amounts of anisotropy[12]. Importantly, past simulations of vapour-deposited glasses have relied on a formation process that involves repeated minimizations of potential energy, which are introduced for computational reasons. As such, past studies have been unable to reveal the role that hot molecules impacting a surface can have on the relaxation of the underlying glassy film. A recent study investigated the formation of highly stable two-dimensional (2D) glasses prepared through a 'pinning' technique[13]. The authors formed equilibrium glasses by freezing in-place a small fraction of the particles in a glass-forming liquid, raising the glass transition temperature above the current temperature, and glassifying the system in an equilibrium configuration. As insightful as the results from the pinning strategy have been, however, such glasses do not incorporate the presence of an interface into the simulations.

Past studies of 2D systems have shed considerable light into the behaviour of glasses. A variety of colloidal particles, including polystyrene and latex, have been shown to assemble into monolayers exhibiting varying degrees of local and long-range order[14,15]. By virtue of being quasi-2D, such studies allow for the direct observation of glassy dynamics, including structural relaxation near the glass transition, thereby serving as a source of validation for theory and simulations[16,17]. Atomic 2D glasses have also been prepared, consisting of silica on a graphene substrate[18,19]. Such systems show a coexistence between crystalline and amorphous regions, which range in size from several unit cells to tens of nanometers across. Going beyond systems of spherical particles, 2D colloidal glasses have been formed using ellipsoids in order resist crystallization[20].

In this work, we build upon these past studies by introducing a PVD formation approach that mimics closely that employed in experiments. Specifically, we avoid the artificial energy minimizations and temperature controls that were employed in past computational studies of 3D systems. Furthermore, by restricting our simulations to 2D systems, where configurations can be more easily visualized and inspected, we arrive at unambiguous correlations between local structure and energetic stability. Three important results emerge from our analysis. First, in contrast to previous reports, we find that vapour deposition leads to glasses whose energetic stability far exceeds that of samples prepared by liquid cooling. Second, it is shown that newly deposited particles generate cascades of hot particles that could serve to relax the interior of the film, and that help explain the advantages of PVD processes for preparation of new glasses. Third, we find that the structure of PVD glasses is isotropic and identical to that of liquid-cooled glasses, provided these two classes of materials are compared under preparation conditions for which their inherent structure energies are comparable.

## Results

**Model system.** The details of the vapour deposition simulations presented here are discussed in the Methods section. Here we point out that the model considered in this study consists of a binary mixture of spheres whose glass-forming behaviour in the bulk has been examined exhaustively, and that vapour-deposited samples are prepared by depositing groups of hot vapour particles onto a substrate held at a temperature $T_s$. Particles are deposited until a desired film thickness of $\sim 35$ molecular diameters is reached. Liquid-cooled samples are prepared by heating vapour-deposited films above $T_g$, and then cooling them at a constant rate to a temperature near zero. A representative system is shown in Fig. 1, where the blue layer at the bottom represents the substrate, the white spheres are of type A, and the black spheres are of type B. Additional sample films are shown in Figs 1 and 2 of Supplementary Information. Vapour-deposited and liquid-cooled films are prepared using a wide range of deposition and cooling rates. The inherent structure energy $E_{IS}$ of a configuration, used to

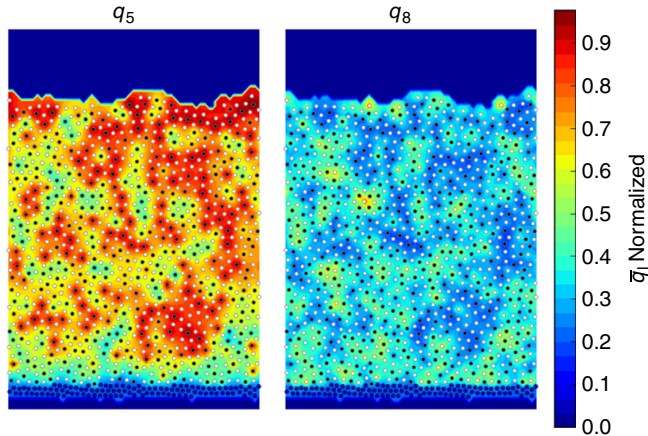

**Figure 1 | Liquid-cooled film where the q5 and q8 are shown for each atom.** This film was cooled with a rate of $t_{cool} = 1.4 \times 10^1 \tau_\alpha$. Type A and B atoms are shown in white and black, respectively, while substrate atoms are shown in blue. This film has an inherent structural energy, $E_{IS}$, of $-3.90$. The background colouring in the left and right panels represents values of bond order parameters $q_5$ and $q_8$ as discussed in the structural features section. Substrate atoms are held tightly in place once equilibrated using harmonic springs. Atoms are kept inside the simulation box using a harmonic repulsive wall as described in Methods.

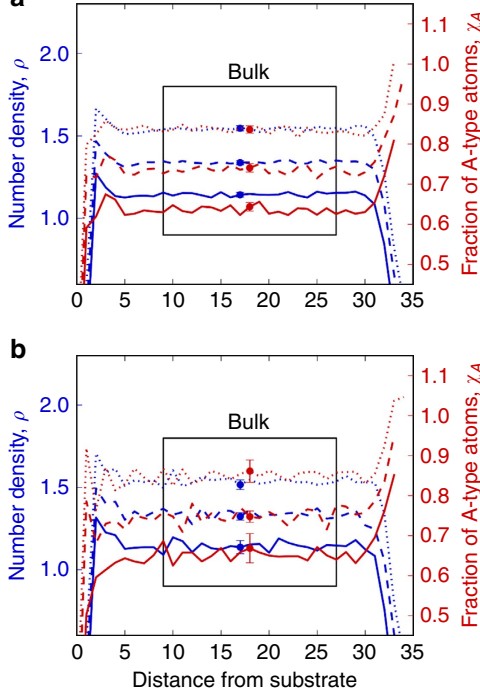

**Figure 2 | Number density and composition for liquid-cooled and vapour-deposited films formed under several conditions.** Data for liquid-cooled films are shown in **a** while data for vapour-deposited films are shown in **b**. The dotted, dashed and solid lines represent films formed with $t = 1.4 \times 10^{(1,2,3)}$ at film temperatures of $(0.75, 0.85, 0.85)T_g$. From top to bottom in each figure, $\rho$ is offset by $(0.4, 0.2, 0.0)$ and $\chi_A$ is offset by $(0.2, 0.1, 0.0)$. In **a**, $t$ refers to $t_{cool}$ and $T$ refers to the film's current temperature in the course of cooling. In **b**, $t$ refers to $t_{dep}$ and $T$ refers to substrate temperature. Only atoms in the bulk region shown are used in calculations unless otherwise specified. We define the bulk region to be several $\sigma_{AA}$ away from where bulk composition and density properties are reached to ensure that edge effects are not present in the data. Error bars represent 95% confidence intervals.

quantify its stability, is the potential energy of a configuration brought to its local energy minimum.

The 2D model considered here exhibits considerable local structure; to quantify this structure, we rely on two bond order parameters that assign values to each particle based on the configuration of its neighbours[21]. The first, denoted by $q_5$, selects for local pentagonal order. The second, $q_8$, selects for local rectangular order. The background colours in Fig. 1 correspond to the magnitude of such order parameters.

**Energetic properties**. The energetic properties of PVD glasses are determined using only particles in the 'bulk' region of the films, which is highlighted in Fig. 2. It corresponds to a wide domain of constant density and composition. Fig. 2 shows results for a variety of PVD and liquid-cooled films. From Fig. 2, we point out two features that arise at the surface of these films: first, the density near the surface decreases gradually. This results from the surface being uneven, as density is simply taken as the number density at a horizontal cross section. Second, $\chi_A$, the mole fraction of type A, rises near the surface of the films, as shown by previously Shi *et al.*[22]. More stable configurations maximize A–B interactions, as $\epsilon_{AB}$ is larger than $\epsilon_{AA}$ and $\epsilon_{BB}$. Type A particles, which are more abundant at $\chi_A = 65\%$, segregate to the surface to maximize these interactions.

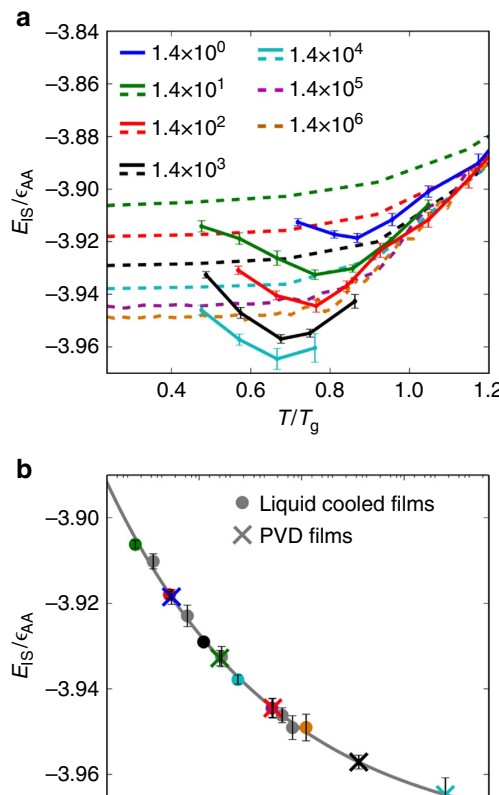

**Figure 3 | $E_{IS}$ of PVD and liquid-cooled films along with liquid-cooling rates predicted to form films with $E_{IS}$ equal to that of PVD films.** (**a**) Inherent structure energy of PVD and liquid-cooled films versus temperature. Dashed lines represent liquid-cooled data while solid lines represent PVD data. For liquid-cooled samples, the film's temperature refers to the temperature at which $E_{IS}$ was calculated during its linear cooling. For PVD films, temperature refers to the substrate temperature with which the film was formed. Legend values refer to $t_{cool}$ or $t_{dep}$ for a given data set, in units of $\tau_\alpha$ (calculated at $T = 1.10T_g$ Supplementary Fig. 3). The ideal substrate temperature decreases as $t_{dep}$ increases for PVD films (Supplementary Table 1). Error bars represent 95% confidence intervals. (**b**) Inherent structure energies of liquid-cooled films at $T = 0.25T_g$ versus $t_{cool}$ with power law fit from equation 1. Colours of the points correspond to the same cooling or deposition rates as in **a**. If a point is grey, that particular cooling rate is not shown in **a**. Ninety-five per cent confidence intervals are shown. The X's represent predicted $t_{cool}$ values necessary to form liquid-cooled films with energy equal to PVD films, as calculated using equation 1. PVD film energies in **b** correspond to that of the substrate temperature that yields optimal stability for each $t_{dep}$.

The inherent structure energy, $E_{IS}$, is an effective measure of the position of a glass on the potential energy landscape[23]. Inherent structure energies of several liquid-cooled and PVD films are shown in panel (**a**) of Fig. 3. The deposition time for vapour-deposited films, $t_{dep}$, corresponds to the interval between addition of new groups of particles to the growing film. During this time, newly deposited particles are allowed to cool down and become integrated into the growing film. The cooling time, $t_{cool}$, is the time over which an ordinary film is cooled from $T = 5T_g$ to $T = 0.2T_g$. Cooling and deposition times are expressed in units of the alpha relaxation time of this system, $\tau_\alpha$, which is calculated using the self-intermediate scattering function at $T = 1.10T_g$ (Supplementary Fig. 3). For all simulations, new, 'hot' particles

are introduced into the system with an initial temperature of $T_i = 5.0 T_g$. The simulated bulk $T_g$ for this material is ~0.21 in Lennard-Jones units, as determined by taking the fictive temperature of a liquid-cooled film prepared with $t_{cool} = 1.4 \times 10^3 \tau_\alpha$.

Previous experimental work has shown that the optimal substrate temperature, $T_s$, for the formation of glasses via PVD lies in the vicinity of $0.85 T_g$ (refs [3,8,24,25]). For the 2D model system considered here, we find that that the optimal substrate temperature (that leading to the lowest inherent structure energy) for a given deposition time decreases as deposition slows. PVD samples formed with $t_{dep} = 1.4 \times 10^0$ show an optimal $T_s$ of $0.87 T_g$, while samples formed with $t_{dep} = 1.4 \times 10^4 \tau_\alpha$ show an optimal $T_s$ of $0.68 T_g$ of $T_g$ (Supplementary Table 1). Furthermore, PVD samples prepared at lower deposition rates exhibit significantly lower inherent-structure energies than those prepared at faster rates. As can be appreciated in Fig. 3, depositing with $t_{dep} = 1.4 \times 10^4 \tau_\alpha$ and $T_s = 0.68 T_g$ gives $E_{IS} = -3.965$ while $t_{dep} = 1.4 \times 10^0 \tau_\alpha$ and $T_s = 0.87 T_g$ gives $E_{IS} = -3.918$. Optimal temperatures are found by fitting a cubic spline to the values of $E_{IS}$ versus $T_s$ in panel (**a**) Fig. 3 and taking the temperature at the minimum energy value.

We suggest that the ideal deposition temperature decreases with slower deposition rate due to a competition between thermodynamics and kinetics. As the substrate temperature decreases, lower energy states become more thermodynamically favourable, but the kinetics to reach such states become slower. As films are formed through more gradual deposition, atoms are allowed more time to approach equilibrium energy states. As originally proposed by Swallen et al.[3], the ideal substrate temperature is where an ideal trade-off is found between which states the system is moving towards (thermodynamics) and how closely the system can approach those states (kinetics).

Panel (**b**) in Fig. 3 shows $E_{IS}$ of liquid-cooled films evaluated at $T = 0.25 T_g$ as a function of cooling time ($t_{cool}$). Previous work on 3D models suggests that $E_{IS}$ varies linearly with $\log(t_{cool})$[11,26]. The 2D glass model considered here exhibits a nonlinear dependence. As shown in panel (**b**) of Fig. 3, a power-law fit of the form:

$$E_{IS} = 0.090 \, t_{cool}^{-0.087} - 3.98 \qquad (1)$$

describes our results reasonably well. Equation 1 can be used to estimate how slowly a liquid should be cooled to form ordinary glass films having inherent structure energies comparable to those of PVD films. These estimated cooling rates are shown by crosses in panel (**b**) of Fig. 3, for $t_{dep}$ values ranging from $1.4 \times 10^0$ to $1.4 \times 10^4$, separated by order-of-magnitude intervals. On the basis of this simple extrapolation, one can anticipate the most stable PVD configuration prepared here to be equivalent to a liquid-cooled sample prepared with $t_{cool} = 1.6 \times 10^{10} \tau_\alpha$, which is $1.1 \times 10^5$ times longer than the time utilized for the slowest cooling rate that we could accomplish with our computational resources.

As PVD films are formed more slowly, the inherent structure energy apparently approaches that of the deepest minima in the amorphous region of the potential energy landscape. By setting the liquid cooling time equal to infinity in equation 1, one can estimate that these lowest energy states have inherent structure energies of $-3.98$. By this prediction, the most stable configurations produced here for $t_{dep} = 1.4 \times 10^4$ with $T_s = 0.67 T_g$ are only 0.013 above this value. We emphasize here that these estimates should be viewed with some skepticism, as the curve shown in the inset of Fig. 3 extends well beyond the data that can be generated with available computational resources. Also note that the more stable vapour-deposited films show a similar, slowing rate of change for inherent structure energy as a function of deposition time, which we believe supports the idea that these

films are gradually approaching the bottom of the amorphous regions of the potential energy landscape.

While the overall composition of each film is fixed, the local composition of the bulk region cannot be controlled precisely. On average, type A particles are excluded from the bulk, and the degree of exclusion varies by film formation type and formation time. It has been shown that $E_{IS}$ for 3D $Ni_{80}P_{20}$ films depends linearly on composition over a small range[26]. That linear dependence is also observed in our 2D films. To account for the variation in $E_{IS}$ due to composition effects, we perform linear fits of $E_{IS}$ to $\chi_A$ for several cooling times. We find $\partial E_{IS}/\partial \chi_A = 1.6$ near $\chi_A = 0.65$ fits well across a wide range of film formation times during both liquid cooling and vapour deposition. The energy of all films is thus interpolated to $\chi_A = 0.65$ for all films, including those used in Fig. 3. The average $\chi_A$ values for PVD and liquid-cooled films in the bulk are 0.648 and 0.637, respectively.

While the aim of this work is to investigate how vapour deposition may influence the structure of glass films, it is worth pointing out that for situations where PVD films and liquid-cooled films exhibit comparable structures, vapour deposition provides an efficient computational method for generating low-energy glasses. For instance, forming a liquid-cooled film with $t_{cool} = 1.4 \times 10^5 \tau_\alpha$ requires $5.0 \times 10^7$ time units and $5.0 \times 10^5$ s on a particular machine. To form a vapour-deposited film of equal energy, one can deposit with $t_{dep} = 1.4 \times 10^2 \tau_\alpha$ and $T_s = 0.76 T_g$, which requires $5.12 \times 10^6$ time units and $4.1 \times 10^4$ s on the same machine, or approximately one order of magnitude less computational (central processing unit, or CPU) time. Using predicted equivalent cooling rates from Table 1 in the Supplementary Information, we anticipate that this difference becomes greater for more stable, lower-energy films. We estimate that our most stable PVD films, prepared with $t_{dep} = 1.4 \times 10^4 \tau_\alpha$, would require over three orders of magnitude more CPU time if prepared by liquid cooling.

**Kinetic properties.** The stability of the PVD films prepared here, based upon two measures, is comparable to that observed in experiment. First, we calculate the fictive temperature, $T_f$, of several liquid-cooled and PVD films. The fictive temperature is defined as the temperature at which the energy line extrapolated

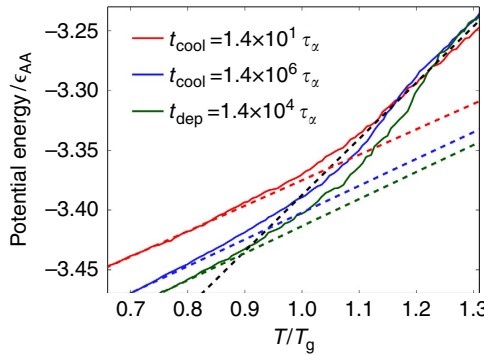

**Figure 4 | Potential energy versus temperature for PVD and liquid-cooled films on heating.** Fictive temperatures, $T_f$, are calculated for three types of films: Shown in red and blue are films formed by liquid cooling at our smallest and largest cooling time, respectively. Shown in green are films formed by vapour deposition at our largest deposition time. The fictive temperatue is calculated to be the temperature where the extrapolated liquid line (dashed black) meets the extrapolated glass lines (dashed red, blue, green). Films are heated from below $T_g$ at a constant rate of $2 \times 10^{-6}$ in reduced units. We calculate fictive temperatures of $1.05 T_g$ and $0.94 T_g$ for the liquid cooled films, and $0.89 T_g$ for the PVD films.

from the glass phase meets the energy line extrapolated from the equilibrium liquid phase, as shown in Fig. 4. In the experiments of Swallen *et al.*[3], the fictive temperature of the glass former 1,3-bis-(1-naphthyl)-5-(2-naphthyl)benzene ($T_g = 347\,K$) was measured for three types of films: ordinary liquid-cooled films, aged liquid-cooled films, and PVD films. These authors found the $T_f$ of these films to be $0.99T_g$, $0.95T_g$, and $0.91T_g$, respectively. Later work in which PVD films were formed at slower deposition rates yielded 1,3-bis-(1-naphthyl)-5-(2-naphthyl)benzene films with $T_f$ of $0.88T_g$ (ref. [27]). Following their work, we calculate $T_f$ for three types of films: films formed by liquid-cooling with $t_{cool} = 1.4 \times 10^1$ $\tau_\alpha$, films formed by liquid-cooling with $t_{cool} = 1.4 \times 10^6$ $\tau_\alpha$ (analogous to an aged glass prepared by liquid cooling), and films formed by vapour deposition using our slowest deposition rate, $t_{dep} = 1.4 \times 10^4$ $\tau_\alpha$. The results are shown in Fig. 4. We find $T_f = 1.05T_g$, $0.94T_g$ and $0.89T_g$ for the three classes of films, respectively. To measure $T_f$, films were heated at a constant rate of $2 \times 10^{-6}$ from well below $T_g$. The ordering and spread of the corresponding fictive temperatures from simulations are consistent with those found in experiment.

Second, we calculate transformation times for both liquid-cooled and PVD films and compare them to experiment. The transformation time is defined as the time required for a material to melt after rapid heating to a temperature above $T_g$. Ultrastable PVD glasses have been shown to melt through a liquid front that originates at the surface of the film. Growth front velocities for ultrastable indomethacin (IMC) have been measured across a wide range of temperatures above $T_g$. These velocities have been found to be constant over a wide range of film thicknesses[28]. We measure film transformation times by rapidly heating films from below $T_g$ to $1.1T_g$, and determining the time required for the film to reach an equilibrium energy, as described in the Methods section. The results, normalized by $\tau_\alpha$ at $T = 1.1T_g$, are shown in Fig. 4 in the Supplementary Information. Energies used to calculate these transformation times are shown in Fig. 5 of the Supplementary Information. The experimental $\tau_\alpha$ of IMC at $T = 1.1T_g$ is $1.3 \times 10^{-4}$ s, while our 2D system shows a $\tau_\alpha$ of $1.48 \times 10^{-10}$ s assuming a Ni-P model. Our most stable PVD films show a transformation time of $158\ \tau_\alpha$, and are 8.93 nm thick, using a Ni-P model. Using data from the literature, we calculate that a 8.93 nm thick film of IMC would melt over $354\ \tau_\alpha$, where $\tau_\alpha$ is measured at $1.1T_g$ for IMC[28]. By this comparison, our PVD films are just over half as stable as would be expected experimentally for films of this thickness. Note, however, that this comparison is highly speculative, given that both the materials and dimensionality of these two types of films are different. We suggest that the lower stability observed in simulations relative to experiment is expected, given that our slowest film growth rate (using a Ni-P model) is 48 μm per second. Experimental growth rates are typically a few nanometers per second, that is, several orders of magnitude slower. Additional details on the conversion to real units and film growth rates are given in the Methods.

**Comparison with 3D films.** Vapour deposition in two dimensions is more efficient than in three dimensions. Two-dimensional films exhibit surface regions which show higher mobility than 3D films assembled using comparable models. This trait allows 2D materials to explore configuration space more effectively, which we suggest leads to the lower inherent structure energy seen in 2D. To compare 2D and 3D films formed by PVD, we examine 3D films with the same interaction parameters as in 2D, but with $\chi_A = 0.80$, as in previous work[11,26]. We define the efficiency of vapour deposition as the ratio of a PVD film's growth rate to the film's equivalent liquid cooling rate. In 2D, equivalent $t_{cool}$ values are found using the power law shown in equation 1. In 3D, $E_{IS}$ is linearly fit to $\log(t_{cool})$ for accessible cooling rates. By combining results from 3D films generated using constant N, V and E deposition (Supplementary Fig. 6) with the 2D data presented here, we estimate that vapour deposition in 2D is between $6 \times 10^1$ and $6 \times 10^2$ times more efficient than in 3D for the films with the lowest inherent structure energies.

Molecules near the surface of a glassy film are more mobile than those in the bulk[29]. Highly mobile molecules can explore configurations more rapidly, thereby allowing films prepared by vapour deposition to reach lower energies than those without mobile surface regions. Consistent with this understanding of surface mobility and our estimated efficiencies, we find that molecules near the surface of 2D films are both more mobile and encompass a thicker region than in 3D. To quantify these observations, we calculate $\langle \Delta r^2 \rangle$ of 2D and 3D films for a range of temperatures and film stabilities. For 2D and 3D samples held at $T = 0.75T_g$, we find that molecules in the surface region are, on average, 70% more mobile than those in the bulk. The high-mobility region extends nearly twice as far into the film than in 3D, as shown in Fig. 5. Surface mobilities do not depend strongly on film stability (Supplementary Fig. 7), though mobilities do depend on film temperature (Supplementary Fig. 8) and particle type (Supplementary Figs 9 and 10). Mechanistically, we suggest that the thicker and more mobile surface layer in 2D allows atoms to sample more configurations before being frozen into their glassy states, thereby enabling exploration of lower energy basins along the free energy landscape.

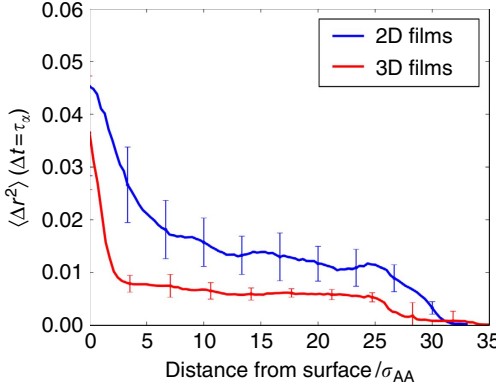

**Figure 5 | Mobility of atoms in both 2D and 3D PVD films.** We measure $\langle \Delta r^2 \rangle$ with respect to distance from film surface calculated over $\tau_{\alpha,2D}$ time units for 2D and 3D films. Both films were formed with $t_{dep} = 1.4 \times 10^1 \tau_{\alpha,2D}$, which gives nearly equal film growth rates. The films are held at $T = 0.75T_g$. Comparing 2D to 3D, the surface region is 70% more mobile and nearly twice as thick in 2D. The surface region is defined using the distance from surface where linear interpolations of the bulk region and the more steeply sloped surface region meet. Error bars represent the standard error from 20 2D and 3D films.

**Heat transfer through films.** As hot vapour particles impact the surface of growing films, energy is transferred from the vapour into the film. In this material, heat transfers along tightly coupled strings of particles. Correlated strings of particles in glasses have been reported before[30]. Note, however, that the strings discussed here are inherently different as they correspond to events initiated by newly deposited hot surface particles that introduce a disturbance. Several representative configurations of long strings are shown in Fig. 6. Particles in these thin strings reach kinetic energies near that of the vapour particle at impact. While 75% of these strings penetrate <4 atom diameters into the film, occasionally, such strings can be significantly longer. In 3% of the

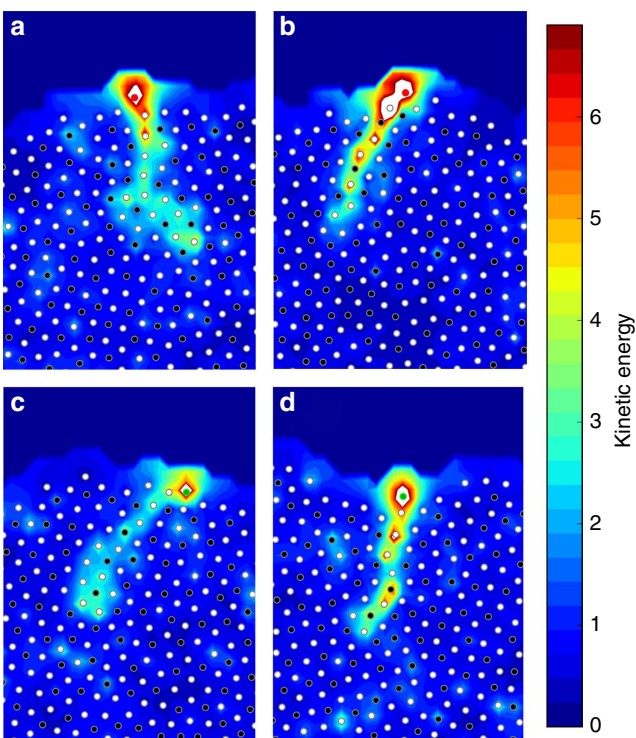

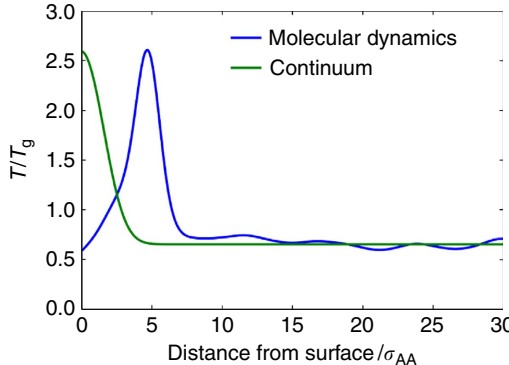

**Figure 6 | Strings of high-energy particles resulting from the impact of a vapour atom during the PVD process.** The four, (**a**–**d**) show independent examples of energy transfer along strings of particles after a vapour particle impacts the surface of the film. The kinetic energy of each particle is normalized by $k_B T_g$. Before impact, the films were equilibrated at $T = 0.5T_g$. As energy travels through the string, it is localized to only one or two atoms at a time. For clarity, atoms involved in a string are shown with their maximum kinetic energy over the lifetime of the string. The particle that impacted the surface is coloured red or green, depending on whether it is of type A or B, respectively. Particles already in the film are coloured white or black for type A or B, respectively.

**Figure 7 | Temperature profiles from continuum and molecular dynamics heat transfer when a vapour particle impacts on the surface of a film.** The temperature profile of molecular dynamics simulations shown in Fig. 6 is shown $2.6 \times 10^{-4}\tau_\alpha$ after the impact of a vapour atom, as compared with temperature profile from similar continuum simulation. The continuum simulation is initialized with a high temperature at its surface to match heat added by vapour atoms' impact. Molecular dynamics simulations that show long strings are used to show the process's effect on thermal transport. The molecular dynamics temperature profile is taken from a narrow slice of the film around the four strings shown in Fig. 6, such that the temperature increase can be easily resolved.

cases, strings penetrate over seven atom diameters into the film, thereby providing a highly focused energy transfer process down to a relatively large depth.

Heat transferred along strings enters the film much more rapidly than would be expected from a diffusive mechanism. To illustrate the difference, one can rely on a simple one-dimensional continuum model where heat only transfers by diffusion. The continuum model's surface is initialized at a high temperature, such that the total amount of heat added to the continuum and molecular dynamics models are the same. Parameters for the continuum model, such as heat capacity and thermal diffusivity, are determined from molecular dynamics simulations as described in the Methods section. One can then generate temperature profiles with respect to distance from the film's surface of these two models as they evolve in time. Figure 7 shows the temperature profile of the PVD films shown in Fig. 6 as compared with the continuum model at $1.1 \times 10^{-2}\tau_\alpha$ after impact or initialization. If one looks at heat transfer averaged over many films, the continuum results are recovered (Supplementary Fig. 11). However, in the case of long strings, heat transfer is much faster and energy is much more localized than in the continuum case, as shown in Fig. 7.

**Structural features**. The 2D films considered here exhibit considerable local pentagonal and rectangular order. Figs 1 and

10 show representative configurations of the system. The $q_5$ and $q_8$ order parameters (which select for local pentagonal and rectangular order, respectively), are used here to analyse the structure of the films[21]. Additional details on the order parameters' selectivity for different geometries are given in Figs 12–14 of the Supplementary Information. The $q_l$ order parameter, which is calculated for each particle based on the arrangement of its neighbours, is defined in equation 2, where $a$ is a particle, $N$ is the set of $a$'s neighbours, and $Y_{lm}$ is the spherical harmonic for the specified $l$ and $m$:

$$q_l(a) = \sqrt{\frac{4\pi}{2l+1} \sum_{m=-l}^{m \le l} |q_{lm}(a)|^2} \tag{2}$$

$$q_{lm}(a) = \frac{1}{\bar{N}} \sum_{n \in N} Y_{lm}(a) \tag{3}$$

High $q_5$ pentagons tend to form mostly as five white type A particles surrounding a single black type B particle. For this reason, $q_5$ is calculated only for type B particles. The $q_8$ parameter is calculated for all atoms. The nearest four neighbours of atoms in high $q_8$ rectangular structures tend to be of different type, thereby maximizing the A–B interaction. Figure 1 shows a contour map of $\overline{q_5}$ and $\overline{q_8}$ values calculated for a liquid-cooled film with a cooling time of $t_{cool} = 1.4 \times 10^1$. Here $\overline{q_l}$ denotes a time averaged $q_l$ parameter averages over in-cage vibrations, as defined in equation 5 in Methods. It can be seen that high-$q_8$ clusters are of medium size, while locally ordered $q_5$ clusters, which cannot tessellate, appear to be pentagonal. A similar coexistence of medium-range ordered clusters and locally ordered structures was reported in a simulated atomic glass system in which particles' anisotropy frustrated crystallization[31].

To assess the extent of order in these films, particle groups are classified as highly ordered or not using a simple cutoff scheme described in Methods. High-order cutoff values are chosen to be $\phi_5 = 0.55$ and $\phi_8 = 22.0$, or 78 and 34% of their values relative to perfectly pentagonal or square configurations (which yield the maximum values for these order parameters). All results can be

reproduced using different cutoffs as shown in Figs 15–20 of Supplementary Information.

We define the degree of order, $D_l$, as the fraction of particles involved in high-$l$ ordered groups. We plot the $D_l$ for all PVD and liquid-cooled films in Fig. 8. We find that as the films become more stable, the $q_8$ character decreases, while the $q_5$ character increases. This can be appreciated by visually comparing Figs 3–8, and by comparing the relatively unstable film in Fig. 1 to the relatively stable films in Fig. 10. Given the direct relationship between these parameters and $E_{IS}$, we conclude that the structure and stability of these films are well captured by the $q_5$ and $q_8$ parameters.

Figure 9 shows the degree of $q_5$ and $q_8$ order versus $E_{IS}$ for all liquid-cooled and PVD films. Only data from films well in the glassy state, $T < 0.2$, are included. The $q_5$ and $q_8$ trends with temperature are similar and independent of the process of formation. These results can in fact be used to estimate inherent structure energy from degree of order since both $D_5$ and $D_8$ behave monotonically with $E_{IS}$. The degree of $q_8$ order for PVD films on average lies slightly below that of liquid-cooled films. We attribute this slight difference to the differences in composition between PVD and liquid-cooled films:

on average, their bulk compositions are $\overline{\chi_A} = 0.648$, 0.637, respectively.

Note, however, that more subtle differences could in principle exist between PVD and liquid-cooled samples. Figure 10 compares vapour-deposited and liquid samples with $E_{IS} \approx -3.95$. The contour map shows no systematic differences in high-order cluster size, location or shape. We find that the size of high-order clusters dependly only on $E_{IS}$ as well, not formation method (Supplementary Fig. 21). Radial distribution functions and structure factors are also calculated for liquid-cooled and PVD films of equal energy, and we find no systemic differences between the two (Supplementary Figs 22–29). Comparing the film in Fig. 1 to the more stable films in Fig. 10, one can appreciate the increase in $q_5$ and the corresponding decrease in $q_8$ character that comes with increasing stability.

To conclude, a new method was introduced to prepare glasses in silico through a process of vapour deposition. The method was applied to investigate a model 2D glass forming liquid. After comparing the structure and energy of the resulting materials to that of ordinary liquid-cooled glassy films, it was found that *in-silico* PVD greatly expands the range of film properties and structures that can be accessed as compared with traditional liquid cooling. In the 2D materials studied here, the range of structures includes pentagonal clusters and square-grid ordered regions of varying size. Under appropriate conditions, forming films by PVD creates extremely low energy films, equivalent to liquid-cooled films cooled five orders of magnitude slower than possible on available computers. By varying the rate of vapour deposition, it is found that the ideal substrate temperature decreases with slowing deposition rate. In 2D, the surface layer of glassy films is thicker than it is in 3D, leading to a more effective PVD formation mechanism. Upon impacting a growing PVD film, newly deposited molecules form strings of hot particles that can reach well into the interior of the film, possibly providing an additional relaxation mechanism that helps the system explore its energy free landscape. An analysis using bond order parameters that select for square and pentagonal order revealed that films transition from a high square-grid character structure to a locally ordered pentagonal structure as

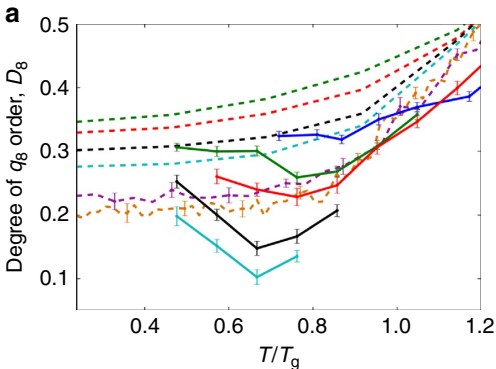

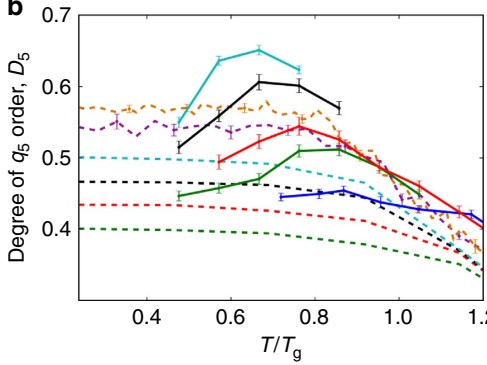

**Figure 8 | Degree of $q_5$ and $q_8$ order in PVD and liquid-cooled films.** Dashed lines represent data from liquid-cooled films, while solid lines represent data from PVD films. **a** Shows data for the $q_8$ order parameter while **b** shows data for the $q_5$ order parameter. The colours correspond to the same rates as in Fig. 3, where blue is $t = 1.4 \times 10^0 \, \tau_\alpha$, orange is $t = 1.4 \times 10^6 \tau_\alpha$, and colours in between are separated by one order of magnitude in cooling rate. $D_5$ increases with film stability while $D_8$ decreases. These data show the same trends as the inherent structure energy shown in Fig. 3, suggesting that these metrics provide a quantitative link between structure and stability in these glassy films. $D_8$ is calculated using all particles in the bulk, while $D_5$ is calculated using only type B particles in the bulk, as pentagonal structures form almost exclusively around these atoms. Error bars represent the standard error; they are only shown for liquid-cooled samples when the error is non-negligible.

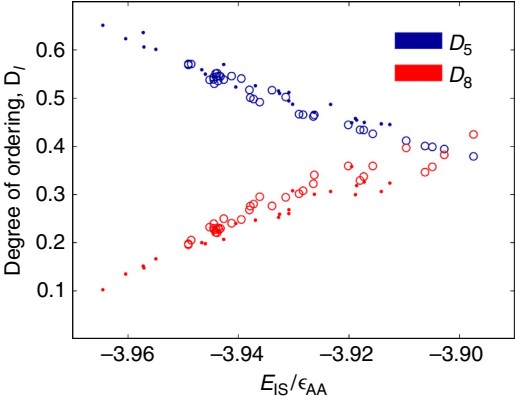

**Figure 9 | Degree of $q_5$ and $q_8$ ordering for vapour-deposited and liquid-cooled films versus inherent structural energy.** Solid circles represent vapour-deposited data while open circles represent liquid-cooled data. Data for liquid cooling is taken from runs with $t_{cool}$ ranging from $1.4 \times 10^1 \tau_\alpha$ and $1.4 \times 10^6 \tau_\alpha$, while data for vapour deposition is taken from runs with $t_{dep}$ ranging from $1.4 \times 10^0 \tau_\alpha$ to $1.4 \times 10^4 \tau_\alpha$. Only data from films with $T < 0.5 T_g$ are used. $q_5$ and $q_8$ show an inverse relationship with $q_5$ increasing with film stability and $q_8$ decreasing. The $q_l$ values of films with equal energy appear substantially equivalent regardless of film formation style, considering that compositions of the two types of films are not identical.

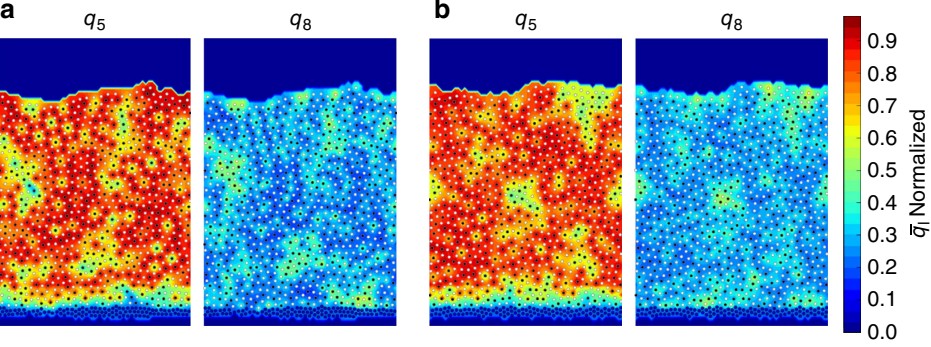

**Figure 10 | Contour maps of $q_5$ and $q_8$ for liquid-cooled and PVD films both with $E_{IS} = -3.95$. a** Shows liquid-cooled film formed with $t_{cool} = 1.4 \times 10^5 \tau_\alpha$ at $T = 0.25 T_g$. **b** Shows vapour-deposited film formed with $t_{dep} = 1.4 \times 10^3 \tau_\alpha$ and $T_s = 0.75 T_g$. These films are of equal inherent structural energy, allowing for direct comparison of the structures. The ordering within these two films shows no systemic differences, suggesting that isotropic PVD glasses are equivalent to those formed by liquid cooling when the films are of equal inherent structure energy.

films stabilize. By examining the change in $D_5$ and $D_8$ in films formed using both methods, it was possible to establish that the degree of order does not depend on the formation type. More generally, the results presented in this work serve to demonstrate that, for the simple, isotropic model considered here, the glassy materials prepared by PVD are the same as those prepared by gradual cooling from the liquid phase, and that PVD glasses correspond to liquid-cooled glasses prepared at extremely slow cooling rates.

## Methods

**Simulation parameters.** The films in this work consist of a binary mixture of Lennard-Jones particles with a third particle type acting as the substrate. The interaction potential is given by equation 4, where $r$ is the distance between two particles, $r_c$ is the distance beyond which interactions are not considered, and $\epsilon$ and $\sigma$ change the strength and range of the interactions.

$$E = 4\epsilon \left( \left( \frac{\sigma}{r} \right)^{12} - \left( \frac{\sigma}{r} \right)^6 \right) \quad r < r_c \tag{4}$$

These simulations use the values $r_c = 2.5$, $\epsilon_{AA} = 1.0$, $\epsilon_{AB} = 1.5$, $\epsilon_{BB} = 0.5$, $\sigma_{AA} = 1.0$, $\sigma_{AB} = 0.8$, $\sigma_{BB} = 0.88$. Values of $\epsilon$ and $\sigma$ for the A and B particles acting on the substrate are 1.0 and 0.75, respectively. The masses of all particles are set to 1.0. The simulation box uses periodic boundary conditions in the $x$ dimension and finite in $y$. The $x$ dimension is parallel to the substrate while the $y$ dimension is perpendicular. The simulations box is $30\sigma_{AA}$ wide and the height is set so that the boundary is $10\sigma_{AA}$ above the surface of the film as it grows. A timestep of $\Delta t = 0.005$ is used for all simulations. A Nosé-Hoover thermostat is used to maintain the temperature of all canonical (constant N, V, and T) ensembles[32].

Inherent structural energies were calculated by minimizing configurations using the FIRE algorithm with energy and force tolerances of $1 \times 10^{-10}$ (ref. 33). All simulations were performed using LAMMPS[34] and all figures were generated using Matplotlib[35].

**Formation of PVD Films.** Vapour-deposited films are formed by initializing a substrate, then adding groups of atoms to the simulation box and allowing them to settle and cool on the growing film. The substrate is formed such that it does not impose any strong ordering on film. First, substrate particles are randomly placed in a small rectangular area near the bottom of the simulation box. The rectangle spans the width of the box and is $3\sigma_{AA}$ tall. The atoms are tethered to their original positions using harmonic springs with a spring constant $k = 5$. The substrate is then minimized using the FIRE algorithm. The substrate atoms are then re-tethered to their minimized positions using harmonic springs with $k = 1,000$. The initial weak spring ensures that the substrate thickness stays roughly constant during the minimization. Throughout the simulation the temperature of the substrate is held constant using a Nosé-Hoover thermostat in an NVT ensemble as described above. A wall with a harmonic repulsive potential is placed $10\sigma_{AA}$ above the substrate. The wall is moved as the film grows to keep the distance between the film and the wall constant.

The film is grown using the following method: ten particles are initialized in a region $3–5\sigma_{AA}$ above the growing film. The particle types are chosen to keep the film configuration as close to $\chi_a = 0.65$ as possible. The particles are initialized with random velocities at $T = 1.0$, as in previous work[11,26]. The new particles and the growing film are then simulated as an NVE ensemble for $t_{dep}$. The new particles cool by natural heat transfer through the growing film to the substrate. This process

is repeated until the films have a height of $\sim 35\sigma_{AA}$. Our method differs from previous work, where the film and vapour atoms are explicitly thermostatted. We find that this method produces lower energies than that employed in previous work (Supplementary Fig. 30) and that film temperature is well thermostatted by the substate (Supplementary Fig. 31).

In all but films formed with $t_{dep} = 1.4 \times 10^0$, the film temperature was tightly distributed around $T_s$. Film temperatures for those formed with $t_{dep} = 1.4 \times 10^0$ were deposited quickly enough that $T_{film}$ was roughly $0.1 T_g$ higher than $T_s$. In these cases, the actual temperature of the film was used in data.

**Formation of liquid cooled films.** Liquid-cooled films are generated by heating vapour-deposited films to $T = 1.0$, then recooling linearly over the time $t_{cool}$. The wall and substrate spring parameters are not changed during this process. To ensure the independence of each liquid-cooled film, the heated configurations are equilibrated for a random time ranging from 100 to 10,000 time units while at $T = 1.0$. The films are cooled to $T = 0.05$, at which point the inherent structural energy has essentially stopped decreasing.

**Transformation time measurements.** Transformation times are measured by heating a film to $T = 1.1 T_g$ over 100 time units, then setting the thermostat to $T = 1.1 T_g$ and measuring the potential energy of the film as it melts. When a film's potential energy is 90% of the way from its initial energy to its final energy, it is said to be transformed. We find that if the films are instantaneously heated from very low temperatures ($T = 0.25 T_g$) to above $T_g$, the films expand extremely quickly, push off the static substrate, effectively 'jump'. For this reason, we introduce the initial heating step.

**Thermal conductivity measurements.** Parameters for the one-dimensional continuum heat transfer were taken from molecular dynamics simulations. In the model, the equation $\frac{dT}{dt} c_v = q = \kappa \nabla T$ is iterated, where $T$ is temperature, $t$ is time, $c_v$ is heat capacity, and $\kappa$ is thermal conductivity. $c_v$ is determined by heating the systems around in the temperature of interest, and measuring the energy required. Thermal diffusivity is measured using the Green–Kubo relation which relates the auto-correlation of heat flux to thermal diffusivity.

**Order parameters.** We assess the order of the systems using a simple high-order cutoff. High-order cutoff values are chosen to be $\phi_5 = 0.55$ and $\phi_8 = 22.0$, or 78 and 34% of their values relative to perfectly pentagonal or square configurations. These cutoff values are chosen in order to discriminate between ordered and non-ordered configurations. Note, however, that the conclusions can be reproduced using other cutoffs (Supplementary Figs 15–20). To create an order metric independent of in-cage vibrations, we average the order parameter $q_l$ defined in equation 5 over $\tau_\beta$ Here $\tau_\beta$ is taken to be 10 Lennard-Jones time units from the time at which the the self intermediate scattering function at $T = 0.8 T_g$ has decayed to its in-cage plateau (Supplementary Fig. 32). This means that we are time averaging over the positions sampled within each atom's glassy cage.

$$\bar{q}_l(a,t) = \frac{1}{\tau_\beta} \int_{t-\frac{\tau_\beta}{2}}^{t+\frac{\tau_\beta}{2}} q_l(a(t')) dt' \tag{5}$$

Particles are then classified as transiently high-order if the $\bar{q}_l$ parameter is above the cutoff value as shown in equation 6.

$$o_l(a,t) = \begin{cases} 1 & \bar{q}_l(a,t) \geq \phi_l \\ 0 & \bar{q}_l(a,t) < \phi_l \end{cases} \tag{6}$$

Finally, we label the particle as high-order if more than half of the transient high-order values in the averaging window of $\tau_\beta$ are 1. Since the $q_8$ metric is intended to select for larger-scale crystallinity, we mark high $q_8$ particles that appear in small clusters and thin strands as not highly ordered.

When selecting highly ordered $q_8$ clusters, two techniques are used to refine groupings. First, any cluster that is of five or fewer atoms is ignored. Second, we note that multiple $q_8$ clusters are occasionally connected by single-atom-wide chains of $q_8$-ordered atoms. For the purposes of counting cluster size, we would like to separate these clusters, as they are structurally distinct (but still connected). To do this, we remove particles from $q_8$ clusters using the following method: First, we count how many of a given atom's neighbours (within a radius of 1.2) are in a $q_8$ ordered group. Then we look at those ordered neighbour particles and perform the same count. If the sum of all of these ordered neighbours is $<5$, we remove the particle from its ordered group, as the atom is likely part of some thin protrusion or connection. A neighbour cutoff of 1.2 was used for equation 3. This value represents the first minimum in the radial distribution function and gave good contrast for bond order parameter values.

**Conversion to real units.** In order to facilitate comparison to experiment, the Lennard-Jones units used in this work are converted to real units. We cast type A particles into nickel and type B particles into phosphorus. The simulated atom of nickel (type A) has mass and Lennard-Jones parameters of unity; to convert into real units, one only needs the energy, length, and mass by which those parameters were normalized. Dimensional analysis shows that the time unit in simulation is given by $t_{unit} = \sigma\sqrt{m/\epsilon}$, with Lennard-Jones parameters for nickel as $\epsilon = 5.65$ kcal mol$^{-1}$ = 23,640 J mol$^{-1}$, $\sigma = 2.552 \times 10^{-10}$ m, and the mass is $58.69 \times 10^{-3}$ kg mol$^{-1}$ (ref. 36). Dividing the $\epsilon$ and mass by Avogadro's number, we find that the real time unit is $4.021 \times 10^{-13}$ s. Our longest PVD simulations lasted $9.2 \times 10^{10}$ simulation timesteps with $dt = 0.005$ Lennard-Jones time units, which translates into a real time of $1.85 \times 10^{-4}$ s. Films are roughly $35\sigma$, or $8.93 \times 10^{-9}$ meters thick, giving a growth rate of 48 μm per second.

**Data availability.** Data and analysis code are available from the authors upon request.

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

## Acknowledgements

We would like to thank David Rodney for many useful conversations. This work was supported by a DMREF grant NSF-DMR-1234320. Fast GPU-accelerated codes for simulation of glassy materials were developed with support from DOE, Basic Energy Sciences, Materials Research Division, under MICCoM (Midwest Integrated Center for Computational Materials).

## Author contributions

D.R.R. carried out the simulations. D.R.R., I.L., M.D.E. and J.J.d.P. analysed and interpreted the results. D.R.R., M.D.E. and J.J.d.P. concieved and planned the study and wrote the paper.

## Additional information

**Competing financial interests:** The authors declare no competing financial interests.

