## [Peer Review File · Nature Communications]

Reviewers' comments:

Reviewer #1 (Remarks to the Author):

The manuscript "Age and Structure of a Model Vapor-Deposited Glass" by Reid et al presents a very nice study of a model (2D) vapor-deposited glass. In particular the study is the first to truly focus in detail on the role of the interfacial properties in determining glass stability within the context of vapor-deposited "ultra-stable" glasses.

Because the study is so well carried out, I am willing to consider acceptance of the manuscript, but I am concerned with a major feature of the paper. In particular I am suspicious of the degree of stability enhancement the authors claim. The timescales in fig. 3b are expressed in units of τ_{α} measured above " T_g " whatever that number means, so that the 10^{11} value is not meaningful.

It is claimed that the lowest E_{is} is obtained about 10 times faster than ordinary cooling. This is very modest, given that no one would ever prepare a glass configuration in silico by slow cooling from very high T , because the "cooling time" then depends linearly on the chosen initial high T . Here the authors start cooling from $T=1$. and $T_g \sim 0.21$ so most of the cooling time is spent in the high- T liquid. If they spent all this time at the lowest T they would get ordinary glasses that would be much better, and perhaps nearly as good as the deposited films.

So overall I think that the improvement over "finite cooling rate" by a factor 10 is no proof that there is real improvement over ordinary simulations if done in a reasonable way.

I insist that the authors attempt to make more clear the true comparison and differences with laboratory-prepared ultra-stable glasses. For example, using measures from experiments (e.g. $t_{\text{transform}}/t_{\alpha}$ after reheating, or other measures) how does stability compare with that seen in the laboratory? Is the difference entirely due to the limitations of the computer? My sense is somehow there is something missing in simulated ultra-stable vapor deposited glasses when compared to what is seen in experiments that is deeper than just finite length/time scale issue of simulations. This question is fundamentally important for understanding the very nature of what imbues vapor deposited glasses with their stability-and thus should also be central to this work.

Reviewer #2 (Remarks to the Author):

In this interesting study, the authors investigate computationally the formation of vapor-deposited glasses, which, under appropriate conditions, has been shown to lead to so-called "ultrastable" glasses, and is hence of considerable scientific and technical interest. Although the manuscript includes several original and potentially significant results, there are numerous issues that must first be addressed by the authors before this work's suitability for publication can be ascertained.

1. The authors provide no physical justification for considering two-dimensional film growth. While the computational advantage is clear, the case needs to be made as to the relevance of this rather artificial set-up to real vapor-deposited glasses, which are invariably "three-dimensional" in the terminology used in this work.
2. The term "onset temperature" (Introduction, first paragraph) should be defined. Non-specialist readers will not be familiar with this terminology.

3. The authors mention, without proper referencing or attribution, that newly deposited molecules can freely explore configurational space near the surface of the growing film (Introduction, second paragraph). The authors should mention some of the numerous papers where this idea has already been expressed.
4. The surface enrichment in A particles in thin films of this particular binary mixture has been reported before (Shi et al., JCP, 134, 114524, 2011). The first paragraph of the Results section should reference that work.
5. The authors should explain why the deposition temperature used throughout is so high (5 times the bulk glass transition), and put this choice in proper experimental context
6. On pages 3 (bottom) and 4 (top) the authors point to a deposition time of 3.3×10^5 in reference to Figure 3. According to the legend in Figure 3, no such case was considered by the authors, although they did consider a cooling time of 3.3×10^5 .
7. The results shown in Figure 3 are very interesting. However, I was surprised not to find a discussion of the physical origin underlying the optimum (minimum) in the vapor-deposited curves of inherent structure energy vs substrate temperature.
8. The term "quasicrystal" has a precise meaning and definition. Instead, the authors use the term to denote particle arrangements that look "almost crystalline". They should use a different terminology, to avoid confusion with the concept of a quasicrystal.
9. The authors should mention the crystal structures that yield maximum values of q_5 and q_8 , and provide appropriate literature references.
10. In choosing the cut-off values of the order parameters q_5 and q_8 above which particles are classified as highly ordered, the authors simply state that above 78 and 34% of the respective maximum values, configurations appear "highly pentagonal or square." This seems highly subjective. What is a "highly pentagonal" configuration? How do the results depend on the choice of cut-offs?
11. If I understood the discussion correctly, the characteristic time over which a particle is classified as high-order (or not) is that corresponding to the decay of the self-intermediate scattering function. Was this time determined by the condition self-intermediate scattering function = $1/e$? self-intermediate scattering function = 0? Or was the time determined from an integral of the self-intermediate scattering function? This should be clarified, as these estimates of the relaxation time can be very different in the supercooled region?
12. Why was the composition of the 3-D films considered by the authors (page 5) different from the 2-D case?
13. On page 8, the authors mention the system's free energy landscape. I believe that the correct terminology here should be energy landscape.
14. The caption to Figure 2 states that the top figure refers to liquid-cooled films and the bottom one to vapor-deposited films. But it is also stated that in the top figure the time refers to deposition time and in the bottom figure to cooling time. The above two statements are contradictory.

Reviewer #3 (Remarks to the Author):

The work represents a continuation of the main authors' efforts to understand the ultradense state created in vapor-deposited glasses. As such, it is an important addition to the ongoing studies of glasses in general and of the dense glasses in particular. The only thing I see missing in the present study is some sense of the fictive temperature of the present simulated glasses and how it compares to, e.g., the glass transition temperature of the glass in the range of the simulated rates and how the conventional glass transition at cooling rates more like 1K/s might compare. Providing a ratio of temperatures may be useful, but the non-conventionality of that type of representation makes it hard to understand exactly where, relative to experiment, the simulations stand. Otherwise, this is an excellent work and should be published.

REVIEWERS' COMMENTS:

Reviewer #1 (Remarks to the Author):

I believe the authors have satisfactorily addressed my concerns. I am willing to recommend publication at this stage.

Reviewer #2 (Remarks to the Author):

The authors have done a very conscientious and thorough job in addressing the comments of the three referees. I believe that in the process the manuscript has gained significantly in clarity. I am pleased to recommend publication of this very interesting and high-quality work.

I wish to add a small clarification: my original comment number 7 did not mean to imply that I was surprised by the appearance of an optimum substrate temperature (Figure 3). I was simply expressing my surprise that the original version of the manuscript contained no discussion of the author's understanding of the underlying mechanism.

Referee 1

The manuscript “Age and Structure of a Model Vapor-Deposited Glass” by Reid et al presents a very nice study of a model (2D) vapor-deposited glass. In particular the study is the first to truly focus in detail on the role of the interfacial properties in determining glass stability within the context of vapor-deposited “ultra-stable” glasses. Because the study is so well carried out, I am willing to consider acceptance of the manuscript, but I am concerned with a major feature of the paper. In particular I am suspicious of the degree of stability enhancement the authors claim. The timescales in fig. 3b are expressed in units of τ_α measured above “ T_g ” whatever that number means, so that the 10^{11} value is not meaningful.

Following the Referee’s suggestion, in order to allow for a more straightforward comparison to experiment, we express time in units of τ_α . τ_α is taken to be the time at which the self-intermediate scattering function has decayed to $1/e$ while at a temperature of $1.10 T_g$. While we are interested in the films below T_g , τ_α increases rapidly as samples approach T_g from above. At $1.10 T_g$, τ_α is equal to 370 Lennard-Jones time units, as explained in the text. All of the times (in units of τ_α) can be converted to Lennard-Jones time units, or to real units for any given system, using the α relaxation time. In response to the Referee’s comment, we have added additional details about how the α relaxation time was measured in the main text, and have added a figure showing the self-intermediate scattering function of these systems at both 10% and 25% above T_g . We would also like to point out that the temperature at which we measure τ_α has changed from $1.15 T_g$ to 1.10 in the revised text. Experimental data for transformation times were available for $1.10 T_g$, but not $1.15 T_g$, and, in order to accommodate the Referee’s suggestion and have only one normalizing timescale, we decided to change all times to be normalized by this new τ_α .

It is claimed that the lowest E_{is} is obtained about 10 times faster than ordinary cooling. This is very modest, given that no one would ever prepare a glass configuration in silico by slow cooling from very high T , because the “cooling time” then depends linearly on the chosen initial high T . Here the authors start cooling from $T=1$. and $T_g \sim 0.21$ so most of the cooling time is spent in the high- T liquid. If they spent all this time at the lowest T they would get ordinary glasses that would be much better, and perhaps nearly as good as the deposited films.

So overall I think that the improvement over “finite cooling rate” by a factor 10 is no proof that there is real improvement over ordinary simulations if done in a reasonable way.

We agree with the Referee in that an improvement by a factor of 10 would not be particularly useful or impressive. The factor of 10, however, is only for the most stable films that can be formed (within a reasonable amount of

computer time) by both vapor deposition and liquid cooling. Importantly, we emphasize that by vapor deposition we can reach low inherent structure energies that are simply not accessible by liquid cooling (at least not within several years of simulation). In that case we must rely on extrapolations from results for accessible cooling rates to hypothetical, lower (but inaccessible) cooling rates, to estimate how long it would take to prepare a liquid-cooled glass. As can be appreciated from the results reported in Table 1, vapor deposition becomes more effective (relative to liquid cooling) as the films become more stable (and exhibit lower inherent structure energies).

Indeed, the most direct albeit imperfect way of calculating the improvement for ultra-stable vapor deposited films is to estimate the cooling time needed to form them by relying on extrapolations on the basis of Equation 1. Using this cooling time for our most stable PVD films, we calculate that it would require three orders of magnitude more CPU time to prepare these films by liquid cooling than by vapor deposition, as clearly stated in the revised text. Given that our most stable vapor-deposited glasses required approximately six weeks of computer time, we would need over 100 years of CPU time to prepare equivalent materials by liquid cooling.

The reviewer also rightly points out that when cooling from a temperature of 1.0, much of the time is spent as an equilibrium liquid. Our coolings were from $T=1$ to $T=0.05$ (Note that $T=1$ corresponds to roughly $5 T_g$). If we instead cooled well-equilibrated films from $T=0.21$ (the glass transition temperature) to $T=0.05$ with the same cooling rates, we should find statistically identical energies, but the cooling time would be 17% of the original time. Our improvement for the most stable PVD films, then, would be reduced from 1000-fold to 170-fold, which is still quite significant.

I insist that the authors attempt to make more clear the true comparison and differences with laboratory-prepared ultra-stable glasses. For example, using measures from experiments (e.g. $t_{\text{transform}} / t_{\text{alpha}}$ after reheating, or other measures) how does stability compare with that seen in the laboratory? Is the difference entirely due to the limitations of the computer? My sense is somehow there is something missing in simulated ultra-stable vapor deposited glasses when compared to what is seen in experiments that is deeper than just finite length/time scale issue of simulations. This question is fundamentally important for understanding the very nature of what imbues vapor deposited glasses with their stability-and thus should also be central to this work.

The Referee's point is well taken. We have added a new section to the main text on the kinetic stability of the simulated PVD films. We investigate stability through both the fictive temperature of several films, and the transformation time, as the reviewer suggested. Using the fictive temperature measurement, we compare our PVD films to our liquid cooled films, and find that the PVD films T_f is approximately $0.05 * T_g$ lower than that of the most stable liquid-cooled film that we can form. Interestingly, this improvement over liquid cooled films is

nearly the same as that found by Swallen, et al., 2007 in their experiments with IMC and TNB. Later experimental work using slower deposition rates lowest T_f of the PVD films by another $0.03 T_g$ (Dawson et al., 2011). However, using the measured transformation times, we estimate that our PVD films are roughly one half as stable as those formed in experiment. We emphasize that our films are formed several orders of magnitude more rapidly than those in experiment, so a lower stability is expected vis-a-vis the experimental measurements.

Referee 2

We thank the Referee for their thorough feedback, they point out many important changes to be made.

1. **The authors provide no physical justification for considering two-dimensional film growth. While the computational advantage is clear, the case needs to be made as to the relevance of this rather artificial set-up to real vapor-deposited glasses, which are invariably “three-dimensional” in the terminology used in this work.**

We thank the Referee for this comment. There is an ample literature, both experimental and computational, pertaining to the use of two-dimensional systems for study of glassy structure and dynamics. Much of what is known about the structure and dynamics of glasses has been learned in experiments with 2D colloidal particle films (Pieranski, 1980, Chakraborty et al., 1991, Denkov, et al., 1992, Ebert et al., 2008, Zheng et al., 2011). Our revised manuscript now includes a discussion of this literature. Here we also point out that a large body of work exists showing that 2D glasses capture the essential features of their 3D equivalents. 2D glasses such as spin glasses (Nemoto et al., 1995) have shown kinetically trapped behavior expected in real 3D glasses. Detailed work has been done on the differences between the 2D Kob-Andersen model used in this work and its 3D counterpart by Flenner and Szamel (Fundamental differences between glassy dynamics in two and three dimensions, Nat. Comm, 2015). While they found the relaxation behavior to be slightly different, the fundamental glassy properties are present in the 2D model. We would like to note that in that work, they compared 2D and 3D materials at the same absolute temperature, not the same T/T_g , and that the T_g values for 2D and 3D models are quite different (0.21 for 2D, 0.335 for 3D). We suspect that if they had compared the materials at the same T/T_g , they would have looked identical. Work in ultra-stable 2D glasses has been done previously by Hockey et al., who used a pinning method as described in the manuscript. In this work, we show that the 2D model captures features of 3D ultrastable glasses in both experiment and simulation, including

- (a) Kinetically trapped states - energies depend on formation time and method
- (b) Highly stable PVD films - Films formed by PVD are more stable than those formed by liquid cooling over much longer times.
- (c) Kinetically stable PVD films - Data has been added to the manuscript on the kinetic stability of these films, and we find that PVD films show high kinetic stability relative to those formed by liquid cooling.
- (d) A single ideal substrate temperature for each deposition rate as shown both in simulation and experiment (Kearns et al., Journal of Chemical Physics, 2007, Singh, Nature Materials, 2013)

We have also added a section on the kinetic properties of these films with direct comparison to experiment, which addresses the validity of this model. We suggest that these features and those described in literature show that the 2D Kob-Andersen model used here captures the relevant physical of glassy behavior and physical vapor deposition.

2. **The term “onset temperature” (Introduction, first paragraph) should be defined. Non-specialist readers will not be familiar with this terminology.**

The reviewer is correct that ‘onset temperature’ has a specialized meaning. A brief definition has been added, and it is accompanied by a citation, which readers will be able to access for further clarification.

3. **The authors mention, without proper referencing or attribution, that newly deposited molecules can freely explore configurational space near the surface of the growing film (Introduction, second paragraph). The authors should mention some of the numerous papers where this idea has already been expressed.**

Thank you for pointing out this omission. We have added three citations: One, the original 2007 Science paper by Swallen et al., which describes this idea, and two others which describe high surface mobility in glassy films: The study by Yang, et al., (Science, 2010) describing the properties of progressively thinner polystyrene films, and that by Zhu, et. al (Journal of Chemical Physics, 2011) describing surface mobility in organic glasses.

4. **The surface enrichment in A particles in thin films of this particular binary mixture has been reported before (Shi et al., JCP, 134, 114524, 2011). The first paragraph of the Results section should reference that work.**

The suggested citation has been added to the first paragraph of the results section; thank you for the suggestion.

5. **The authors should explain why the deposition temperature used throughout is so high (5 times the bulk glass transition), and put this choice in proper experimental context**

For our deposition temperature (which should not be confused with the substrate temperature), we choose to use the same deposition temperature as that used in previous work (Singh, et al., Nature Materials, 2011, Lyubimov et al., 2013) , $T=1.0$. This value has been used to reproduce the distinguishing characteristics of PVD glass. Experimental deposition temperatures are typically not reported, since substances are typically heated until they vaporize at low pressure. In “Physical vapor deposition as a route to hidden amorphous states“ by Dawson et al., PNAS, 2011, the methods section states “Vapor depositions were carried out in a vacuum chamber with a base pressure of 10^{-7} to 10^{-8} torr. The chamber is designed such that the crucible loaded with crystalline IMC can be positioned 3 cm beneath either a quartz crystal microbalance (QCM; Sycon Instruments) or a silicon wafer. While positioned under the QCM, the crucible was heated by using a resistive wire heater to achieve the desired deposition rate.” The exact source temperature is not reported. This is the case for much experimental literature. In this study with IMC, the source temperature is at least 434K, which they report as the melting temperature for IMC (Indomethacin). They report the T_g as 315 K, giving $T_{dep} \geq 1.38 T_g$.

The source temperature is reported in one article we find: In work with PVD films of toluene, the source temperature was 298 K, while T_g is 117 K (Ahrenberg et al., 2013). Their deposition temperature, then, was roughly $2.5 T_g$. With this deposition temperature, ultrastable glasses were successfully formed. To address what effect adding atoms at $T=1.0$ may have, we show in the supplementary information the measured film temperature during deposition, and find that for all but the fastest deposition rate, the film temperature does not deviate from the substrate temperature.

6. **On pages 3 (bottom) and 4 (top) the authors point to a deposition time of 3.3×10^5 in reference to Figure 3. According to the legend in Figure 3, no such case was considered by the authors, although they did consider a cooling time of 3.3×10^5 .**

The error has been fixed; thank you for the correction.

7. **The results shown in Figure 3 are very interesting. However, I was surprised not to find a discussion of the physical origin underlying the optimum (minimum) in the vapor-deposited curves of inherent structure energy vs substrate temperature.**

We have added an explanation which reflects our understanding of the origin of the ideal substrate temperature. Similar observations are made in the original experiments of Swallen et al., Science, 2007.

8. **The term “quasicrystal” has a precise meaning and definition. Instead, the authors use the term to denote particle arrangements that look “almost crystalline”. They should use a different terminology, to avoid confusion with the concept of a quasicrystal.**

The reviewer is correct. The word quasicrystalline has been replaced by “pentagonal” or “locally ordered.”

9. **The authors should mention the crystal structures that yield maximum values of q5 and q8, and provide appropriate literature references.**

Text describing maximally ordered structures has been added. We have also added an extensive section in the Supplementary Information describing the properties of the q5 and q8 order parameters, showing their selectivity for different geometries and how this selectivity changes as the configurations become disordered. While many references describe maximum values for q4, q6, and q5 in 3D (Bond-orientational order in liquids and glasses, Steinhardt et al., 1983, Bond orientational order in atomic clusters, Chakravarty, 2002, Isothermal-isobaric computer simulations of melting and crystallization of a Lennard-Jones system, Nosé, 2002), to the best of our knowledge references do not exist that state the maximum values of q5 and q8 in 2D. An online source (<http://www.pas.rochester.edu/~wangyt/algorithms/bop/>) provides a value for a 2D hexagonal lattice. We calculate the same value using their configuration.

10. **In choosing the cut-off values of the order parameters q5 and q8 above which particles are classified as highly ordered, the authors simply state that above 78 and 34% of the respective maximum values, configurations appear “highly pentagonal or square.” This seems highly subjective. What is a “highly pentagonal” configuration? How do the results depend on the choice of cut-offs?**

We have clarified in the revised manuscript what structures these values are in reference to. The reviewer makes an excellent point, that the cut-offs chosen are ultimately arbitrary. We have added to the supplementary information several figures calculating our crystallinity metrics (the data shown in Figs 9 and 10), but using two different cutoff values. The q8 cutoffs used are 20 and 24 (where 22 is used in the main text), and, for q5, 0.50 and 0.60 (where 0.55 is used in the main text). These changes in

the cutoff values are enough to increase or decrease the calculated fraction of crystallinity significantly. The results shown in Figure 10 hold for both of the new cutoff values. We have also added to the Supplementary Information sample configurations which represent a range of q5 and q8 values. In addition to these added configurations, one can also appreciate what we call a ‘highly ordered’ structure by inspecting Figure 1, where films are shown with a heat map or their q5 and q8 values overlaid on top of each other. Note that the q5 parameter is calculated only for Type-2 atoms (colored black in Figure 1), so the heat map values on the white Type-1 atoms are simply interpolated from the nearest Type-2 atoms and are not meaningful, as explained in the text.

11. **If I understood the discussion correctly, the characteristic time over which a particle is classified as high-order (or not) is that corresponding to the decay of the self-intermediate scattering function. Was this time determined by the condition self-intermediate scattering function = 1/e? self-intermediate scattering function = 0? Or was the time determined from an integral of the self-intermediate scattering function? This should be clarified, as these estimates of the relaxation time can be very different in the supercooled region?**

The characteristic time here is the beta relaxation time, which represents the time when particles have equilibrated within their local “caged” environment. This is taken to be when the self-intermediate scattering function has to its caged plateau value of roughly 0.9. We find this time to be 10 Lennard-Jones time units at 0.8 Tg. This has been clarified in the revised text. The purpose of this time is to provide a timescale over which one can average to find the average position of a particle in its “caged” environment. As long as this time is at least 10 Lennard Jones time units, and significantly less than tau alpha, the results are unchanged.

12. **Why was the composition of the 3-D films considered by the authors (page 5) different from the 2-D case?**

Compositions of 80-20 in 3D and 65-35 in 2D are widely used, standard compositions for the Kob-Andersen model. These compositions have been shown to resist crystallization well and promote glass formation. For example the 80-20 model has been used in 3D in Stringlike Cooperative Motion in a Supercooled Liquid, Donati et al., 1997, and Dynamical Heterogeneities in a Supercooled Lennard-Jones Liquid, Kob et al., 1997, while the 65-35 model has been used in 2D in Adam-Gibbs Relation for Glass-Forming Liquids in Two, Three, and Four Dimensions, Sengupta et al., 2012, Fundamental differences between glassy dynamics in two and three dimensions, Flenner et al., 2015. To the best of our knowledge,

using the 80-20 mixture in 2D produces large HCP regions of only type A, while using 65-35 in 3D gives large FCC regions of A and B, so these compositions are not practical for studies of the properties of glasses.

13. **On page 8, the authors mention the system’s free energy landscape. I believe that the correct terminology here should be energy landscape.**

We have changed our wording to “energy landscape”.

14. **The caption to Figure 2 states that the top figure refers to liquid-cooled films and the bottom one to vapor-deposited films. But it is also stated that in the top figure the time refers to deposition time and in the bottom figure to cooling time. The above two statements are contradictory.**

The caption has been fixed; thank you for the correction.

Referee 3

The work represents a continuation of the main authors' efforts to understand the ultradense state created in vapor-deposited glasses. As such, it is an important addition to the ongoing studies of glasses in general and of the dense glasses in particular. The only thing I see missing in the present study is some sense of the fictive temperature of the present simulated glasses and how it compares to, e.g., the glass transition temperature of the glass in the range of the simulated rates and how the conventional glass transition at cooling rates more like 1K/s might compare. Providing a ratio of temperatures may be useful, but the non-conventionality of that type of representation makes it hard to understand exactly where, relative to experiment, the simulations stand. Otherwise, this is an excellent work and should be published.

Thank you for the feedback. We have added a section to the main text on the kinetic stability of the simulated PVD films. We investigate stability with both the fictive temperature of several films, and using the transformation time, as the Referee suggested. Using the fictive temperature measurement, we compare our PVD films to our liquid cooled films, and find that the PVD films' T_f is roughly $0.05 T_g$ lower than that of the most stable liquid cooled film we form. Interestingly, this improvement over liquid cooled films is nearly the same as is found experimentally by Swallen, et al., 2007 in IMC and TNB. Using the measured transformation times, we estimate that our PVD films are roughly one half as stable as those formed in experiment. Given that our films are formed several orders of magnitude more quickly than those in experiment, such a lower stability is to be expected. These new findings have been included in the revised text.

We would also like to point out that the temperature at which we measure tau alpha has changed from $1.15 T_g$ to 1.10 in the revised text. Experimental data for transformation times was available for $1.10 T_g$, but not $1.15 T_g$, and in order to have only one normalizing timescale, we decided to change all times to be normalized by our new tau alpha.